# Wireless Home Assistive System for Severely Disabled People

**Chung-Min Wu [1], Yeou-Jiunn Chen [2,3], Shih-Chung Chen [2,\*] and Chia-Hong Yeng [2]**

[1]    Department of Intelligent Robotics Engineering, Kun Shan University, Tainan 710303, Taiwan;
      cmwu@mail.ksu.edu.tw

[2]    Department of Electrical Engineering, Southern Taiwan University of Science and Technology,
      Tainan 71005, Taiwan; chenyj@stust.edu.tw (Y.-J.C.); steve10624@gmail.com (C.-H.Y.)

[3]    Allied AI Biomedical Research Center, Tainan 71005, Taiwan

\*    Correspondence: chung@stust.edu.tw; Tel.: +886-6-253-3131 (ext. 3335)

**Abstract:** A lot of people with severe disabilities such as amyotrophic lateral sclerosis, motor neuron diseases, cerebral palsy, stroke, and spinal cord injury with intubation always have different degrees of communication problems. Therefore, it is very important to develop an effective and easy use assistive communication system for the severely disabled. In this study, a wireless home assistive system (WHAS) with different types of assistive input accessories sensors, Morse code translator, and human machine interface is developed and tested to help the severely disabled communicate with people and machines. A Morse code translator is implemented as an assistive communication core device to facilitate the input of the severely disabled. For the proposed human–machine interfaces, personal computer-based alternative augmentative communication is developed for patients to communicate with other people easily. To promote the quality of life, the home appliance control interface is developed for the severely disabled to directly control the functions of home appliances by themselves. The experimental results showed that the proposed WHAS is practical and feasible. Therefore, the proposed approach can help severely disabled individuals effectively interact with their surroundings.

**Keywords:** amyotrophic lateral sclerosis; motor neuron diseases; cerebral palsy; stroke; spinal cord injury; information communication technology; home automation

## 1. Introduction

The lives of people with severe disabilities such as amyotrophic lateral sclerosis (ALS), motor neuron diseases (MND), cerebral palsy (CP), spinal cord injury (SCI), and stroke with intubation can be effectively improved to some extent with the help of assistive communication systems [1]. When some of the original human functions such as the functions of movement and language are disabled, it would result in very serious problems and inconvenience in the disabled's daily life. According to the world report on disability produced by World Health Organization [2], the proportion of the world's population over 60 years old will increase from 11% to 22%, and 15% of the world's population by 2050 are estimated to be living with a disability. Once the patients lose their capabilities of movement and language, they can not communicate with the world smoothly. The disabled person can not express his/her own opinion conveniently, and then the caregiver can not understand the needs of the disabled person. Therefore, the arguments and mistakes are easily generated between the disabled and the caregiver and result in the disabled's bad mood. The bad relationship between the disabled and the caregiver during long-duration will let the bad things happen which is hard to regret. Hence, it's very

important to develop an assistive communication system for improving the communication capability of the disabled person.

"Communication" is a process of exchanging opinions, transmitting messages, and establishing a relationship. Effective communication could be done through the good medium or tool of communication. The objects of communication not only include humans but also machines. In general, the medium of communication could be language, text, movement, expression and the possible interactive information. A lot of severe disabilities, such as ALS, MNDs, CP, SCI, and stroke, cause people to lose their abilities of speech and movement forever; therefore, they can not communicate with people directly. Of course, they can not control their own daily life by themselves as they wish, either. Gradually, the daily life of the disabled becomes colorless and meaningless.

The severely disabled can mean quadriplegia—that the people with serious diseases or handicaps such as movement and speech disabilities lie in bed year round and can do nothing but very limited functions such as blinking the eyes, twisting cheeks, twitching the finger, etc. What they can do every day is to watch the ceiling and ruminate. The severely disabled neither can move by themselves nor communicate with the others. According to the limits of severe disabled, it is very difficult for the severely disabled to communicate with people or machines. However, the sensory nerves and the autonomic nervous system are unaffected, the severely disabled might maintain the capabilities of hearing, sight, touch, smell, taste, thinking, and cognition. In short, to solve the communication problem is the most urgent issue, and it would make the severely disabled live more meaningfully and happily.

In December 2014, an assistive context-aware toolkit (ACAT) platform was developed by the Intel research team and the famous astrophysicist named Stephen Hawking, who is a patient with ALS. The researchers in the world can free access the related information of ACAT [3] to implement more effective assistive communication systems because Intel has announced the source codes of the program and the related software on the website. Currently, the important functions of Intel ACAT include keyboard simulation, word prediction, speech synthesis, etc. It can help the severely disabled with movement and speech problems control the computer to communicate with the outside world, including editing, managing documents, navigating the web, writing, and accessing emails. Nevertheless, the body conditions of each disabled people are not the same; thus, ACAT might be not always suitable for all the people with ALS or other severe disabilities. For example, some severely disabled subjects are not used to wearing the sensor on the cheek or waiting for a scan for input, either.

Augmentative and alternative communication (AAC) is an alternative way to help people with speech challenges or language disorders find effective and appropriate ways to express their basic needs. Many AAC systems have been implemented to help the disabled in communicating with others or devices [4–7]. Most of the off-the-shelf AAC devices belonging to the standalone system are portable, simple, and convenient, supporting brief communication functions of short messages for the disabled. An AAC device has a communication board with texts and icons that pronounce a speech by touching the command sheets, and it is used to help people with speech disabilities [4]. A tongue drive system is proposed to enable people with severe disabilities to access their environment [5]. The information of discrete breathing patterns is identified to represent the pre-defined words and help the disabled to express their needs [6]. An AAC app was implemented to use speech symbol technology to express their thoughts, needs, and ideas [7]. Unfortunately, these AAC systems mentioned above are only suitable for slightly disabled people with speech disorders but still with movement functions they can control, such as patients with CP or slight stroke. Thus, it is important for the disabled to select a suitable and easy communication interface.

Recently, many studies in areas such as home automation, environmental monitoring, and internet of things (IoT) have been widely promoted to improve the quality of life [8–12]. Many wireless communication technologies are utilized to solve the smart home and home automation control problems. Some papers [13–16] discussed the application of IoT, wireless Bluetooth, and X-10 protocol for the environmental condition monitoring and smart home system for the disabled. In addition to

wireless communication technologies, there are also some papers discussing algorithms, such as the algorithms of different neural network designs and artificial intelligence embedded in the decision mechanism of system control core to make the smart home system more intelligent [17,18]. Moreover, needs differ as the disabilities vary; each disabled person may need different assistive input devices depending on their disabilities. Thence, different command input mediums were discussed, such as the medium of voice-controlled hands-free [19] and the brain–computer interface (BCI) communication pathway [20,21]. Shinde et al. developed an Arduino-based application including an android application, a wireless module, and Arduino devices to control smart appliances [22]. Rathi et al. proposed a hand gesture recognition algorithm for implementing a gesture human–machine interface that can be used to operate all the smart home appliances [23]. Rawat et al. adopted automated speech recognition to help the severely disabled operate computers and home appliances without clicking buttons [24]. Gagan used the Intel Galileo Board to achieve information including temperature, humidity, gas, smoke, motion, and fire, and then the user can control the home appliances [25]. However, most of the related researches are for normal people or the slightly disabled, while there are fewer papers talking about communication and home automation solutions for the severely disabled. In [8], a wireless home automation system is proposed for the deaf, dumb, and the people with Alzheimer's, but the disabled subjects with speech, hearing, and memory problems mentioned that still have normal moving functions can operate the home appliances easily. Therefore, to develop a specific assistive device for home automation, environmental monitoring, or computer input would be a great and important issue for the severely disabled.

In this study, a wireless home assistive system (WHAS), including Morse code translator (MCT) and human–machine interface, is developed to help the severely disabled. To meet the special physical requirements or limitations of the severely disabled, many different types of assistive input accessories such as mechanical switches, sensing switches, and bio-signal switches are adopted. To help the severely disabled input commands and messages, MCT is developed. For the MCT, Morse code is applied to represent the commands in simple form and then Morse code, a fuzzy algorithm based-Morse code recognition, is developed to accurately identify commands. To assist the severely disabled in communicating with humans or machines, human–machine interfaces including personal computer-based alternative augmentative communication (PC-based AAC) and a home appliance control interface are developed. Therefore, the severely disabled can have other optional methods to communicate with others in this WHAS, and then they can do more things alone, even including entertainment in daily life.

The remainder of this paper is organized as follows. Section 2 describes the proposed WHAS, including MCT and the human–machine interface. To evaluate the proposed approach, Section 3 presents the results of a series of experiments, and some discussions are also described. Finally, the conclusions and possible improvements for the future development of this work are discussed in Section 4.

## 2. Materials and Methods

The architecture of WHAS shown in Figure 1 is mainly to introduce the complete concept of the whole assistive communication system, including three important parts. (1) The assistive input accessories include three kinds of digital switches such as the mechanical switch, sensing switch, and bio-signal switch. (2) MCT is an important core device embedded with fuzzy algorithms for translating Morse codes into American standard code for information interchange (ASCII) and playing the roles of keyboard and mouse. Meanwhile, a Bluetooth component is integrated with MCT for the wireless communication function. (3) In addition to the two kinds of hardware mentioned above, the related application software including the PC-based AAC and home appliance control interface were developed to be compatible with the assistive input accessories and MCT. The WHAS consists of three important parts mentioned above in Figure 1 and is described in detail as follows.

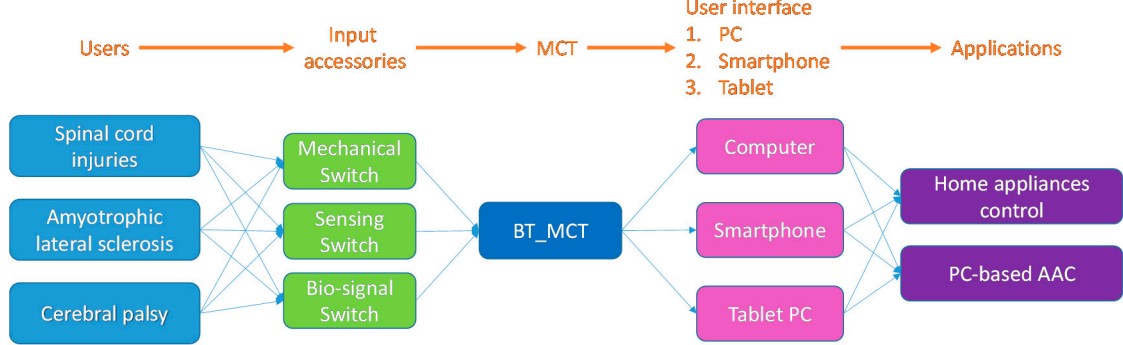

**Figure 1.** The architecture of the wireless home assistive system (WHAS).

### 2.1. The Assistive Input Accessories

In order to help the severely disabled person control the MCT easily, we developed many kinds of assistive input accessories for them to choose, which are shown in Figure 2. When using an input digital switch, the time duration of press-and-hold is adopted to distinguish the long or short tonesof Morse code. In Figure 2, the assistive input accessories can be divided into three types, consisting of mechanical, sensing, and bio-signal (e.g., eye movement). The choice of input accessories is in accordance with the physical disability condition of each user. All three types of assistive input accessories mentioned above can output the similar digital outputs (different time duration of high/low level voltages) to represent the long/short tones and long/short silences of Morse codes to MCT. Any type of digital switches can be accepted as an assistive input accessory in WHAS if it can output similar digital output.

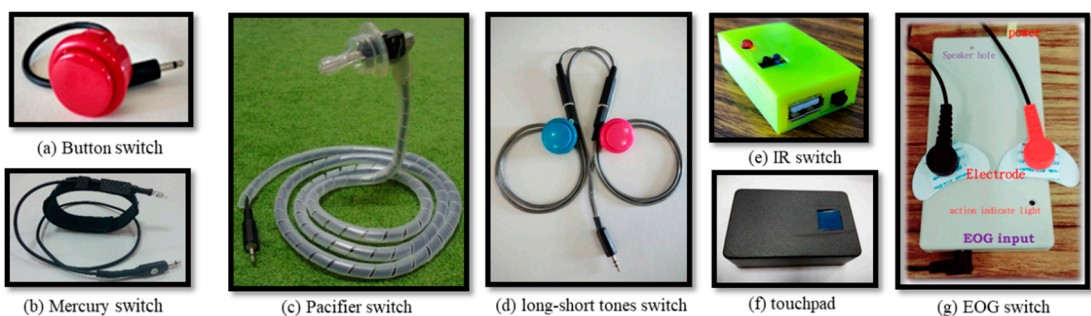

**Figure 2.** The kinds of input accessory. EOG: electrooculography.

The mechanical switches are shown in Figure 2a–d. The mechanical switch is mainly based on a push-button switch and a micro switch, which is not necessary to additionally provide a power to drive the switch. The mechanical switch is connected to a 3.5 mm mono male connector. When the switch is clicked, it will output a low voltage level; otherwise, it will output a high voltage level. To quickly input the Morse code to the MCT, a long–short tones switch (as shown in Figure 2d) is designed to input the long or short tones by using two buttons respectively.

The sensing switches are shown in Figure 2e,f. An operational amplifier (OPA) comparator follows a photo sensor to provide a binary output. The OPA reference voltage value decides the reflection distance of the infrared light detected by the photo sensor. In order to avoid being triggered by unstable infrared reflection signals multiple times, we designed a Schmitt trigger circuit with feedback, whose reference voltage increases the stability of the special infrared switch. At the same time, two types of switches were designed according to different users' requests: one is the positive edge trigger (0→1) switch, and the other is the negative edge trigger (1→0) switch. In addition, a 555 oscillator integrated circuit (IC) is utilized to generate sound feedbacks by an external speaker module to check and confirm the Morse codes inputted when the user inputs Morse codes (short tones

and long tones). The surface mount device type resistors, capacitors, and ICs were combined with a TCRT5000 infrared reflection sensor to reduce the size (4.8 cm × 2.7 cm) and costs. The special IR switch is shown in Figure 2e.

The electrooculography (EOG) assistive input switch shown in Figure 2g is a bio-signal switch. EOG is one of the common effective ways to measure eye movement. The range of EOG voltage is about 50 to 3500 µV, the frequency range is about DC 100 Hz, and the range of eye movement angle is about ±30 degrees, which is proportional to the EOG voltage, so we can identify the direction and angle of the eyeball according to the eye movements. The EOG measurement circuit detects and amplified the differential signal of the eye movement by the INA128 instrumentation amplifier with the magnification 251; then, the second-order 1.6–10 Hz infinite impulse response band pass filter is designed for noise cancellation, which is finally followed by an inverse amplifier with 100 times amplification. After the EOG analog signal measurement, the single Schmitt-trigger inverter (SN74LVC1G14) is utilized to complete analog/digital (A/D) conversion and become an EOG digital switch. Therefore, the user can use the EOG digital switch connected with MCT as a keyboard or mouse to input Morse codes by controlling the user's eye movement alone.

## 2.2. Morse Code Translator (MCT)

The functional block diagram of the MCT is shown in Figure 3. The time duration of press-and-hold from assistive input accessories can be separated into four kinds of base elements of Morse codes, i.e., a long tone, short tone, long silence interval, and short silence interval. The time duration ratio of a long tone to a short tone is 3 to 1, so is the time duration ratio of a long silence interval to a short silence interval. It is difficult for normal people who are not well trained to accurately input Morse codes according to this principle of time duration ratio, not to mention the severely disabled. In order to overcome the problem of time duration ratio control, the fuzzy recognition algorithm is embedded in MCT to automatically adjust the threshold value between a long tone and short tone, and the threshold value between a long silence interval and short silence interval as well according to the on/off intervals controlled by the user respectively, so as to increase the identification accuracy of long and short tone/silence intervals. The Morse code fuzzy recognition algorithm architecture for a single switch is shown in Figure 4. In Figure 4a, the Morse codes are divided into two kinds of sequences, one is tone (on) interval sequence ($n_T(k)$) and the other is silence (off) interval sequence ($n_I(k)$), which are adjusted by two functions ($N_T$ and $N_I$) respectively to simplify the Morse code recognition process. After the signal adjustment, each threshold level ($T_{TK}$, $T_{IK}$) of the corresponding long and short tone/silence intervals are adjusted using the fuzzy algorithm, respectively, to improve the recognition accuracy of long and short tone/silence intervals. $R_{TK}$ and $R_{IK}$ are the identification functions for the long/short tone/silence of Morse codes according to $T_{TK}$ and $T_{IK}$ respectively. Once the fuzzy recognition process for the Morse codes is finished, MCT can translate the inputted Morse codes to readable characters (in ASCII format) or commands and transmit them into a computer, smartphone, or tablet computer via the Bluetooth interface, replacing the keyboard and mouse.

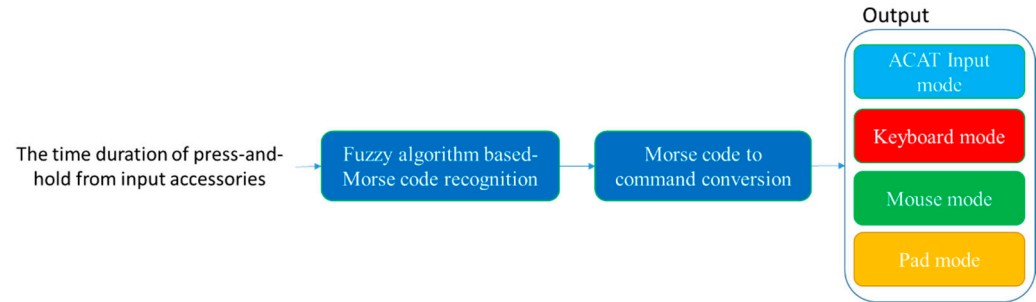

**Figure 3.** Functional block diagram of MCT.

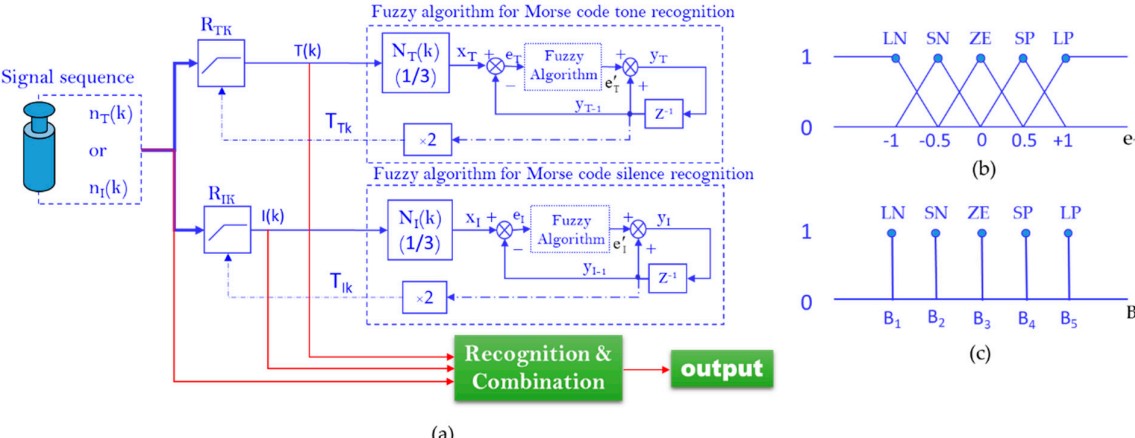

(a)

**Figure 4.** Fuzzy recognition algorithm of MCT for a single switch, (**a**) the flow chart of fuzzy Morse codes recognition in MCT; (**b**) the membership function of fuzzification; (**c**) the membership function of defuzzification.

In Figure 4, two control flowcharts are used for the tone and silence recognition of Morse code. The recognition procedure of Morse code tone sequence in the tone recognition flowchart is described below.

**Step 1.** In order to simplify the Morse code recognition process, we utilized the signal adjustment function $N_T$, i.e., to update $x_T$ according to $T_{TK}$ in Equation (1) recursively.

$$N_T \begin{cases} x_T = n_T(k), & \text{if } n_T(k) < T_{TK} \\ x_T = \frac{1}{3}n_T(k), & \text{if } n_T(k) > T_{TK} \end{cases} \tag{1}$$

where $N_T$ is the signal adjustment function, $x_T$ is the output of $N_T$, $n_T(k)$ is the original input signal, and $T_{TK}$ is the kth threshold of tone to distinguish between long and short elements.

**Step 2.** The prediction error $e_T$ in Equation (2) is an input to the fuzzy algorithm, and it is created by the difference between $x_T$ and $y_{T-1}$,

$$e_T = x_T - y_{T-1}. \tag{2}$$

In the fuzzy algorithm, a linguistic fuzzy rule is utilized to calculate modified error $e'_T$. Five linguistic parameters are negative large (LN), negative small (SN), zero (ZE), positive small (SP), and positive large (LP) (Figure 4b).

The fuzzy rules in conditional expressions (3) are described as "if the conditional premise is true, then the corresponding result is met":

$$\begin{aligned} &\text{IF } e_T \text{ is LN, then } e'_T \text{ is LN.} \\ &\text{IF } e_T \text{ is SN, then } e'_T \text{ is SN.} \\ &\text{IF } e_T \text{ is ZE, then } e'_T \text{ is ZE.} \\ &\text{IF } e_T \text{ is SP, then } e'_T \text{ is SP.} \\ &\text{IF } e_T \text{ is LP, then } e'_T \text{ is LP.} \end{aligned} \tag{3}$$

**Step 3.** The output variable ($e'_T$) of the fuzzy algorithm block is calculated by the center of gravity method and can be defined as Equation (4)

$$e'_T = \frac{\sum\limits_{i=1}^{n} W_i B_i}{\sum\limits_{i=1}^{n} W_i} \tag{4}$$

where $W_i$ is the membership grade e of the i-th premise in the fuzzy inference rules shown as conditional expression (3), and $B_i$ is the center value of the i-th conclusion in the inference rule. The range of the defuzzifier is from $B_1$ to $B_5$ (Figure 4c).

**Step 4.** The predictive output ($y_T$) is revised by $y_{T-1}$ and $e'_T$, as shown in Equation (5)

$$y_T = y_{T-1} + e'_T. \tag{5}$$

$T_{TK}$ is regarded as the threshold value of the recognition function ($R_{TK}$) for the next time, and it is twice of $y_{T-1}$. By repeating steps 1–4, the system can automatically adjust the threshold value ($T_{Tk}$) by following the typing speed variation of the user. Similarly, the same recognition procedure for long or short silence intervals between Morse code tones is finished through the four steps mentioned above.

MCT has four kinds of output modes: ACAT input mode, keyboard mode, mouse mode, and pad mode, respectively. MCT basically adopts the standard code sets established (Appendix A) by the Morse 2000 working group that defines the corresponding Morse codes of the alphanumeric characters, symbols, functional commands, and instructions to switch between different function modes. In ACAT mode, MCT can send a confirm command to select the option on the recursive scanning keyboard interface in the environment of the ACAT system.

In keyboard mode, MCT plays the role of the keyboard that allows the user to input characters, symbols, and functional commands via a single switch. The user inputs appropriate Morse codes according to the keyboard mode table of Appendix A. The fuzzy recognition program inside MCT can identify the corresponding alphanumeric character (e.g., a–z, 0–9), symbol ($,/, … ) or function key (Tab, Caps Lock, Ctrl, Alt, Shift, Enter, … ). Then, this output is transmitted to the computer, smartphone, or tablet via a USB or USB On-The-Go (USB OTG) interface. In addition, MCT can recognize the input of composite keys as well.

In mouse mode, the eight directions of mouse movement, mouse button operations (left click, right click, and double left click), and mouse drag function are shown in the mouse mode table of Appendix A. MCT enables the user to access the full functions of the mouse through the input of four Morse code elements at most.

In pad mode, MCT plays the role of the function keys of a portable device that allow the user to input music player function commands to operate the portable device, such as "home", "play", "stop", "up volume", "down volume", etc.

*2.3. The Hardware of MCT*

The entity of MCT (shown in Figure 5) incorporates eight parts placed on the front side (Figure 5a) and back side (Figure 5b) panel respectively, including (1) two kinds of input interface, (2) microprocessor, (3) eight light-emitting diode (LED) lights, (4) initial typing speed adjustment encoder, (5) power, (6) single/double key input, (7) Human Interface Device (HID) Bluetooth module, and (8) a 3.7 V lithium battery. The microprocessor processes the real-time recognition of Morse codes and converts them to the ASCII codes, and it sends them to the screen of the computer or portable device (tablet PC and smartphone) through the Bluetooth interface. The eight LED lights appearing on the front panel indicate ACAT input (AI) mode, keyboard mode, mouse mode, single/double key input, Caps Lock, Shift, Ctrl, and Alt, respectively. This configuration provides the user with a simple way to confirm the current state of MCT. The initial typing speed adjustment encoder allows modifying the operating speed of the assistive input accessories (i.e., the initial threshold value of the fuzzy algorithm) for users with different degrees of disability; therefore, the MCT can real-time track the typing speed of the user. The power connector is a USB 2.0 micro-B port, it can connect to a PC or mobile power source directly when the lithium battery powers down. The single/double switch input allows the user to choose one or two keys to input Morse codes. The HID Bluetooth module [26] supports the transmission of the command to the computer, tablet PC, and smartphone from the microprocessor.

The lithium battery supports the power of MCT when the user goes outside. The box of MCT (Figure 5c) is designed by a 3D printer and can be adjusted freely according to the user's preferences.

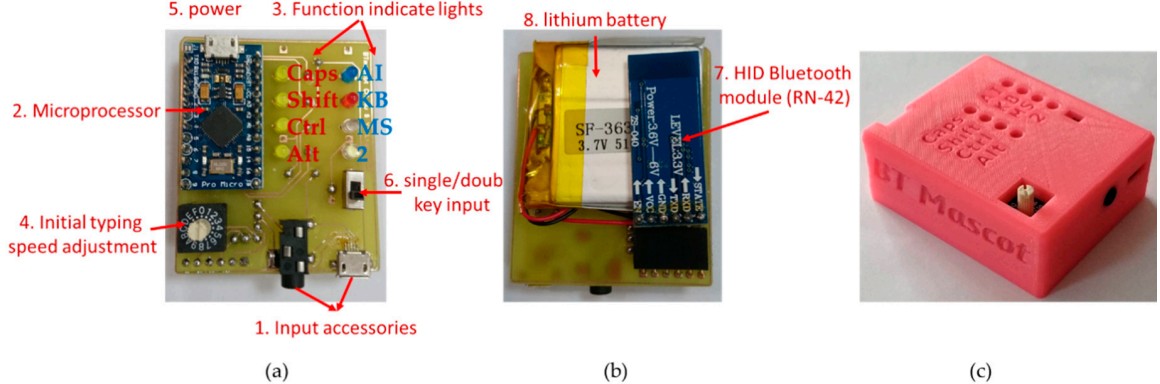

**Figure 5.** The entity of MCT: (**a**) The front side of the panel, (**b**) The back side of the panel, and (**c**) MCT product.

### 2.4. The Human–Machine Interface

This study designs two kinds of human–machine interfaces of MCT for the user: PC-based AAC software and the home appliance control interface. The PC-based AAC software programmed in LabVIEW graphic language includes a big dialog database with words, phrases, or sentences and the corresponding voice database. Basically, the database of PC-based AAC can be classified into 3 kinds of dialogs: the dialog for daily life, the dialog for greetings, and the dialog for emotional expression. It's very easy and flexible to change or update new option of words, phrases, or sentences for the interface of PC-based AAC. In general, the PC-based AAC is quicker and more effective for the severely disabled to reply to the caregiver or family. More and more useful and popular words, phrases, sentences, voices, and even different languages can be appended to the database as long as the hard drive or disk is big enough. The voice database built into the PC-based AAC includes a synthesized computer voice and a real human voice, e.g., the voice of the family or caregiver, or even the real voice of the severely disabled. The PC-based AAC can be operated with the help of MCT; the user can click the option on the interface shown in Figure 6 when MCT is set in mouse mode or typing word in keyboard mode. Compared with the general standalone AAC products for the disabled with speech disorders, the PC-based AAC is easier and it is more feasible to update or increase the scale of the word database or phrase database without hardware limitations.

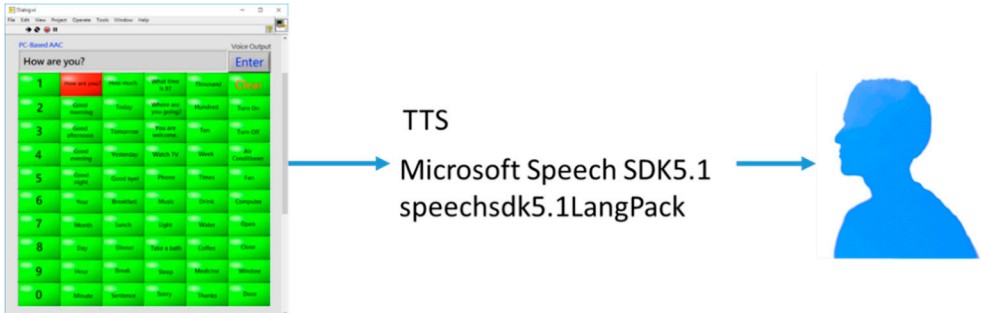

**Figure 6.** The user interface of personal computer-based alternative augmentative communication (PC-based AAC) with voice output included in WHAS.

The home appliance control interface shown in Figure 7 includes three parts of dialogs, which are used for general home appliance control, air conditioner control, and TV channel selection. Besides the software interface mentioned above, there are still a few hardware components such as an infrared

transceiver, Arduino module, Bluetooth module, and a power on/off control box (for controlling general home appliances, e.g., fan, music, light etc.). The user can control the general home appliance, air conditioner, and TV channels through the home appliance control interface by themselves. Two kinds of wireless communication protocols are adopted to finish the wireless home appliance control: one is infrared protocol, and the other is Bluetooth protocol. The user can operate the functions of the home appliance target by using an assistive input accessory and MCT according to the home appliance control interface on the computer screen. After that, the home appliance control system will automatically choose the infrared protocol (for air conditioner, TV channel selector) or Bluetooth protocol (for general home appliances, e.g., fan, music, light, etc.) to communicate with home appliances and complete home appliance control automation.

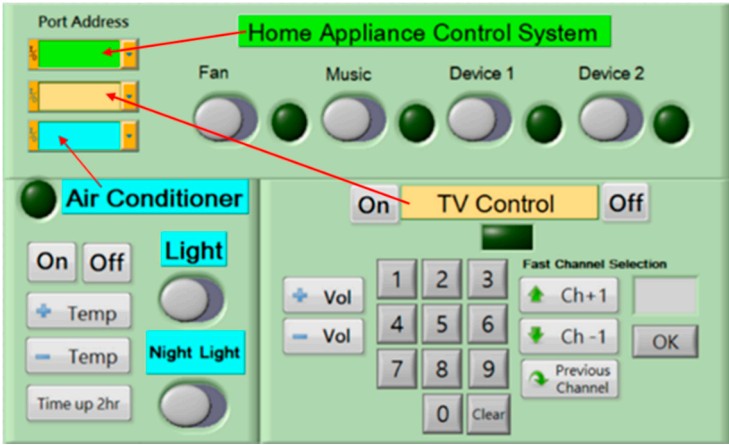

**Figure 7.** The user interface of the home appliance control system for the severely disabled.

## 3. Results and Discussions

There were eight severely disabled subjects who signed the agreements to attend the test of the project under the institutional review board (IRB) project supervision (No. B-ER-105-396 approved by National Cheng Kung University Hospital, Tainan, Taiwan). The degree of the subjects' barriers is shown in Table 1. Six severely disabled subjects are male and they are 52, 47, 41, 41, 39, and 30 years old respectively. Five of them are SCI subjects and one is a CP subject. Two female subjects are 58 and 56 years old with SCI and ALS disease, respectively. In Table 1, C1–C7 represent the cervical spine, which has 7 stacked bones called vertebrae that are labeled C1 through C7. Athetosis is a symptom characterized by slow, involuntary, convoluted, writhing movements of the fingers, hands, toes, and feet and in some cases, arms, legs, neck, and tongue. M and F are male and female, respectively. All of them have been confined to bed for many years. The experimental results of MCT and the related home appliance manipulation are described as follows.

**Table 1.** The degree of the subjects' barriers. ALS: amyotrophic lateral sclerosis.

| Subject | Age | Gender | Injured Area | Mobility | Speaking Ability |
|---------|-----|--------|--------------|----------|------------------|
| 1 | 52 | M | C5–C7 | Finger fretting | Limited speech |
| 2 | 58 | F | C3–C4 | Quadriplegic | Faint |
| 3 | 41 | M | C5–C7 | Finger fretting | Normal |
| 4 | 41 | M | C2–C4 | Quadriplegic | Faint |
| 5 | 39 | M | C3–C4 | Quadriplegic | Faint |
| 6 | 30 | M | C2–C3 | Quadriplegic | Faint |
| 7 | 47 | M | Athetosis | Quadriplegic | Slurred |
| 8 | 56 | F | ALS | Finger fretting | Faint |

## 3.1. The Experimental Results of MCT

In this subsection, the MCT in mouse mode and keyboard mode is evaluated, and the results are detailed in the follows.

### 3.1.1. The Efficiency of MCT used as a Mouse

In order to evaluate the efficiency of MCT with a button used as a mouse, a mouse function test interface for MCT was designed and shown in Figure 8. During the test of MCT set in mouse mode, there are nine icons on the screen (17″, 1920 × 1080) in which the arrangement of icons is shown in Figure 8: one start button is on the center of the screen, four icons numbered 1, 3, 5, and 7 are at the four corners of the screen and four icons numbered 2, 4, 6, and 8 are on the four edges of the screen. The subject was asked to move the cursor to each flower icon position numbered sequentially from 1 to 8 and click each icon by inputting the corresponding Morse codes of mouse control commands to MCT. The software can count and show the total time cost when the user finishes all the procedures per cycle, so that the user can know how fast he/she finishes this mouse mode test of MCT. Each subject is asked to play the mouse function test 3 times; the subject can rest for 3 min between two cycles. The 8 severely disabled participants were invited to attend the mouse function test, and the test result is shown in Figure 9; the average time cost of the first, the second, and the third cycle are 86.45, 74.62, and 67.76 s, respectively.

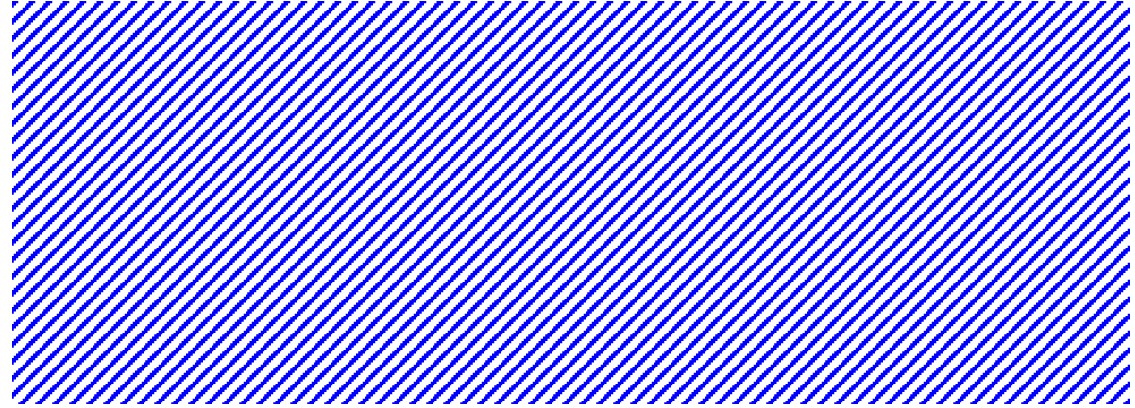

**Figure 8.** A mouse test interface for evaluating the efficiency of MCT (**a**) before the test and (**b**) after the test.

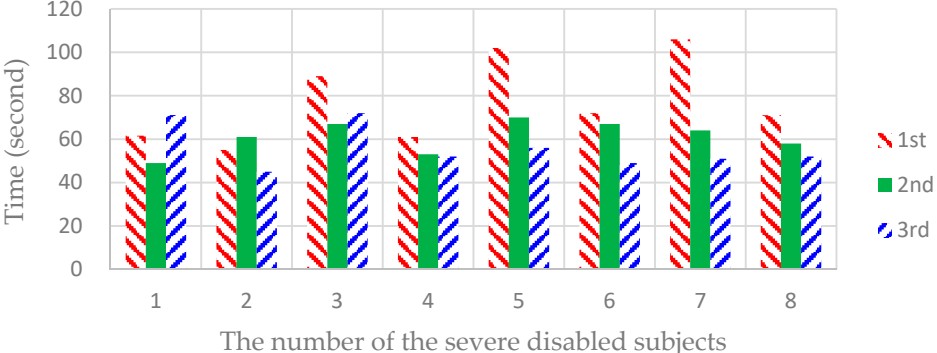

**Figure 9.** The test data of the mouse mode of MCT for 8 severely disabled subjects (3 times per subject).

### 3.1.2. The Efficiency of MCT used as a Keyboard

In order to verify the efficiency of the fuzzy algorithm and keyboard mode performance of MCT, eight severely disabled subjects were asked to type 26 English alphabets ('a' to 'z', sequentially) with the help of a pacifier with a limit switch inside that is regarded as the input accessory connected with

MCT. An example of the tone and silence intervals of the Morse codes of each corresponding alphabet letter in the sequence is shown in Figure 10. For instance, the corresponding Morse codes of characters 'a' and 'b' are shown in Table 2. The analysis of tone and silence intervals in Figure 10 is shown in Table 3.

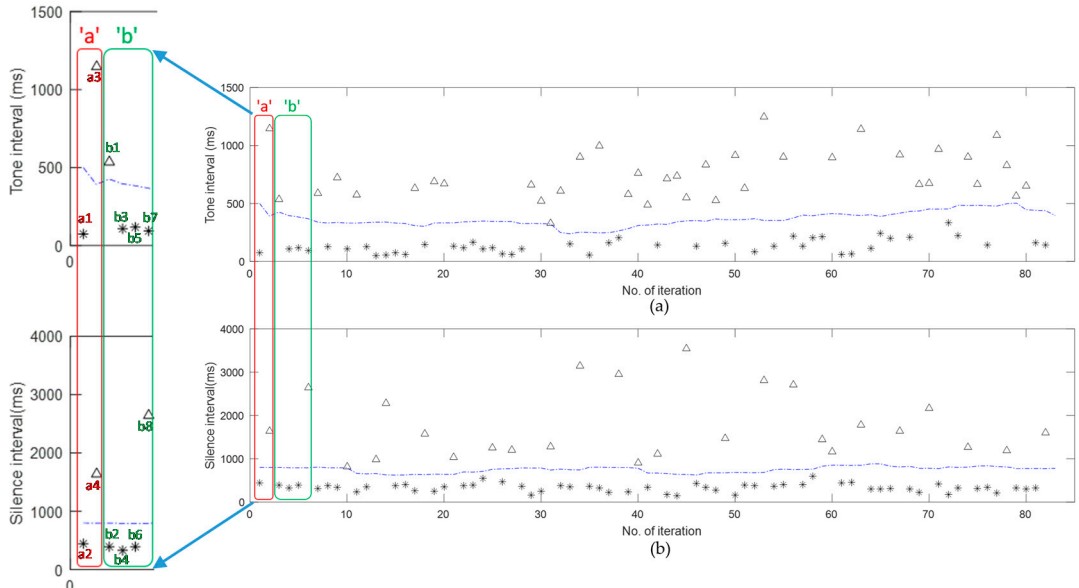

**Figure 10.** The intervals of corresponding Morse codes relating to (**a**) tone sequence of Morse code and (**b**) silence sequence of Morse code. (*: short tone, Δ: long tone).

**Table 2.** The input order and corresponding Morse codes of characters 'a' and 'b'.

| | 'a' | | | | 'b' | | | | | | | |
|---|---|---|---|---|---|---|---|---|---|---|---|---|
| No. | a1 | a2 | a3 | a4 | b1 | b2 | b3 | b4 | b5 | b6 | b7 | b8 |
| Morse code | * | | Δ | | Δ | | * | | * | | * | |
| Tone interval (ms) | 76 | | 1149 | | 535 | | 108 | | 119 | | 93 | |
| Silence interval (ms) | | 444 | | 1642 | | 386 | | 331 | | 398 | | 2641 |

(*: short tone, Δ: long tone).

**Table 3.** Statistical analysis of Morse codes typing (ms).

| | Tone | | Silence | |
|---|---|---|---|---|
| | *Long* | *Short* | *Long* | *Short* |
| *Mean* | 748.6 | 134.4 | 1755.2 | 337.3 |
| *Standard deviation* | 207.6 | 60.4 | 771.9 | 91.7 |

In Figure 10, '*', 'Δ', and dash line (_._.) are used to represent the short tone/short silence, the long tone/long silence, and the variable threshold, respectively. The variable threshold adjusted by the fuzzy algorithm can separate the long and short tone/silence interval clearly. It proves that the fuzzy algorithm of Morse codes can effectively trace the typing speeds of different subjects whether the subject is a green hand or a specialist of Morse code typing. In tone sequences, the mean values of the short tone intervals and long tone intervals of Morse codes typing are 134.4 ± 60.4 and 748.6 ± 207.6 milliseconds (ms), respectively. In silence sequences, the mean values of the short silence intervals and long silence intervals of Morse codes typing are 337.3 ± 91.7 and 1755.2 ± 771.9.6 ms, respectively. The results indicate that the maximum short tone and minimum long tone are 194.8 ms and 541 ms, respectively. The long-to-short ratio of Morse code is about 2.78 smaller than 3; i.e., the standard ratio of long to short Morse code. Thus, the fuzzy algorithm can adjust the threshold value and follow the

typing speed variation of the subject to successfully recognize the corresponding Morse codes of the English alphabet.

According to the test results of eight severely disabled subjects, the average typing speed is about 13.63 characters/minute, the fastest typing speed is 16.17 characters/minute, and the slowest typing speed is about 11.23 characters/minute. The accuracy of the Morse codes typing test with MCT is about 92%, and the errors include insertion errors (1.71%), deletion errors (2.99%), and substitution errors (5.13%). The reason why the errors happened mainly results from the Morse code typing skill and proficiency of the severely disabled subjects, i.e., the severely disabled are not very familiar with all the combination of Morse codes or the use of the accessory connected with MCT. The best way to improve the accuracy of Morse codes typing is to practice more, as practice makes perfect.

Regarding the keyboard mode test of MCT, the threshold value can be auto-adjusted depending on different Morse code input speeds with the help of a fuzzy algorithm. We can ensure that the performance of MCT is satisfied with the severely disabled to help them implement the communication task with a computer.

### 3.2. The Experimental Results of the Human–Machine Interface

In this subsection, the human–machine interfaces, which were implemented as PC-based AAC and home appliances control, were evaluated, and the results are detailed in the following section.

### 3.2.1. The Experimental Results of PC-Based AAC

Generally, the AAC system is designed for people with speech disabilities. Most of the off-the-shelf AAC products are standalone and have their own motherboard system, and they might not be compatible with the other electronic accessories, such as mouse, keyboard, memory card, etc. Most of the AAC end users have speech disabilities but still have limited movement functions to proceed with AAC operation. However, the definition of the severely disabled in this study is a person lying on a bed all the year who has serious speech and movement disabilities simultaneously. Hence, it means that most of the severely disabled are not able to manipulate general standalone AAC products. That is why we developed the new assistive communication system associated with the PC-based AAC function to give more optional communication methods to the severely disabled. However, MCT can replace the ordinary keyboard and mouse to input the corresponding Morse codes of any alphabet or symbol on the keyboard to communicate with the other people with words, phrases, sentences, or even voice output. The speed efficiency of MCT is still worse than PC-based AAC. The PC-based AAC is software programmable and it is easy to upgrade the words or voice database regarding the corresponding icons for the words, phrases, and simple sentences.

### 3.2.2. The Experimental Results of Home Appliances control

There were eight severely disabled subjects attending the research of WHAS. Two cases of severely disabled are shown in Figure 11a,b respectively: one of the severely disabled with SCI was using the functions of home automation to control home appliances, the other severely disabled with CP was using the Intel ACAT platform to input texts. Both of these two cases adopted different MCT-connected assistive input accessories (one adopted the pacifier with a limit switch inside, the other adopted a mechanical button). The related sensors and accessories used in WHAS are shown in Figure 11c.

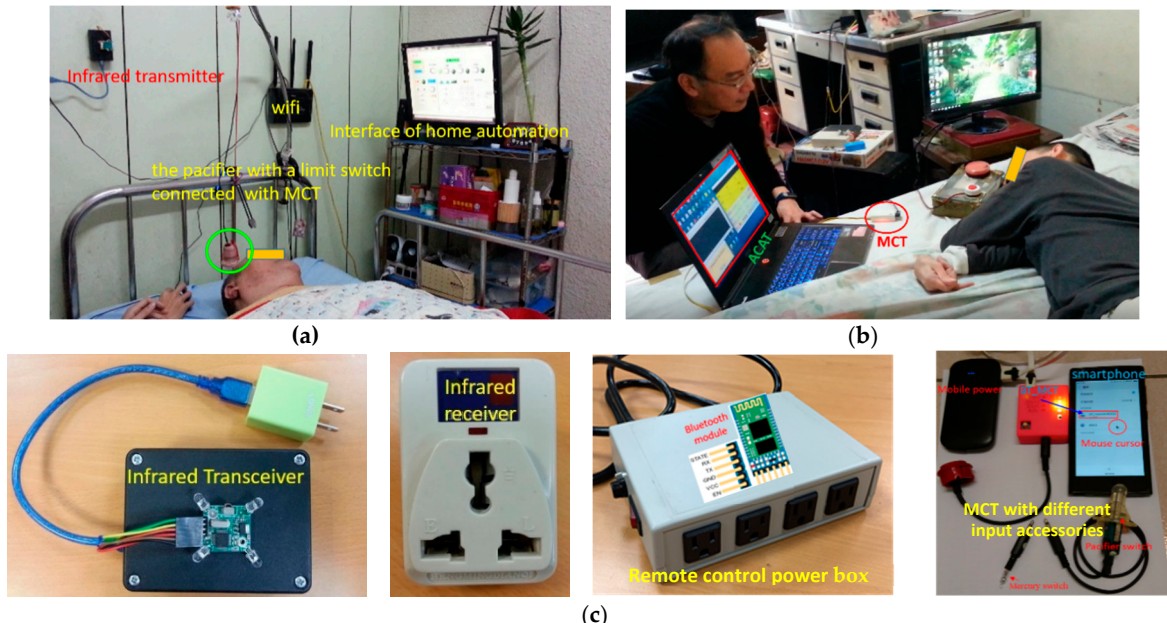

**Figure 11.** The real situations of two severely disabled subjects manipulating home automation and Intel ACAT platform with the help of MCT, (**a**) Home automation; (**b**) Intel ACAT; (**c**) The related hardware (Arduino chip, wirelessly controlled power box, Bluetooth and infrared transceiver, etc.) and assistive input accessories used in WHAS.

### 3.2.3. The Subjective Evaluation of Human–Machine Interface

To subjectively evaluate the performance of the human–machine interface, the mean opinion scores (MOS) are adopted in this subsection. The subjects gave MOS on a scale of 1 to 5, i.e., 5 for excellent level, 4 for good level, 3 for fair level, 2 for poor level, and 1 for unsatisfactory level. In these experiments, the subjects were asked to operate the human–machine interface including PC-based AAC and home appliance control by using the specific input accessories connected with MCT 2 to 3 h every day lasting for two months. After two months, the acceptance of user friendly and the degree of practicality are evaluated, and the results are shown in Table 4. In Table 4, the average scores for the acceptance of user friendly and the degree of practicality are 4.13 and 4.00, respectively. The results indicate that the acceptance of user friendly obtains a higher MOS since the use of MCT can achieve high accuracy and it can prevent frustration in inputting commands or messages. Moreover, the MOS showed a satisfaction degree in using MCT to communicate with others or devices. Thus, the use of MCT can effectively help severely disabled subjects in daily life.

The severely disabled 52-year-old male is regarded as the best example; he got an injury in the army. In the past, he has been confined to bed for 20 years. He can't move his hands, body, or legs, but he can twitch his fingers and still has limited speech abilities. All he could do before he joined this research is watch cable TV or listen to music all day. He had to depend on his family for almost everything. However, he can control the computer to communicate with other people through the Internet by using Line, Skype, and Facebook software and control home appliances such as the TV, air conditioner (AC), music, fan, and light by himself now. The most important reason is that he was well trained to manipulate MCT, the home appliance automation system, and Intel ACAT platform via a specific pacifier with a limit switch inside as an input accessory, as shown in Figure 2c. He can do many things alone in his daily life, including control the computer, light, and fan, select TV channels, control music, etc. He was the best example to demonstrate and promote WHAS.

**Table 4.** The mean opinion scores (MOS) regarding the acceptance of user friendly and the degree of practicality.

| | *No of Subjects* | | | | | | | |
|---|---|---|---|---|---|---|---|---|
| | *1* | *2* | *3* | *4* | *5* | *6* | *7* | *8* |
| *Acceptance of human–machine interface* | 3 | 4 | 4 | 5 | 4 | 4 | 5 | 4 |
| *Degree of practicality* | 3 | 4 | 4 | 4 | 4 | 4 | 5 | 4 |

## 4. Conclusions

In this study, a WHAS including assistive input accessories, MCT, and human–machine interfaces has been successfully developed to help the severely disabled communicate with humans and machines. Many different types of assistive input accessories such as mechanical switches, sensing switches, and bio-signal switches can be effectively used to meet the special physical requirements or limitations of the severely disabled. MCT has been developed to ease the severely disabled in inputting commands and messages. Moreover, human–machine interfaces including PC-based AAC and home appliance control interface are developed to ease the severely disabled in communicating with human or devices, and then the quality of life can be greatly improved. The experimental results showed that the accuracy of the Morse codes typing test with MCT is about 92% and the errors include insertion errors (1.71%), deletion errors (2.99%), and substitution errors (5.13%). Therefore, the efficiency of MCT is also proved to be practical and feasible for the severely disabled. Besides, subjectively evaluating the performance of the human–machine interface showed that the severely disabled are very satisfied with the proposed PC-based AAC and home automation applications. Once the severely disabled can manipulate the computer, Internet, PC-based AAC, and home automation appliances, they will be getting more confident and independent; thus, less help from caregivers or family will be needed. In the future, a multi-language version of the novel assistive communication can be continuously further developed to help more severely disabled in the world.

## 5. Patents

The invention patent certificate No. I576722 relating to MCT was approved by the Intellectual Property Office, Ministry of Economic Affairs, Taiwan.

**Author Contributions:** Conceptualization, C.-M.W. and S.-C.C.; Data curation, C.-M.W., S.-C.C. and C.-H.Y.; Formal analysis, C.-M.W. and Y.-J.C.; Investigation, C.-M.W., Y.-J.C. and S.-C.C.; Methodology, C.-M.W. and Y.-J.C.; Project administration, C.-M.W. and S.-C.C.; Resources, C.-M.W. and C.-H.Y.; Software, C.-M.W.; Supervision, C.-M.W. and S.-C.C.; Validation, C.-M.W., Y.-J.C. and S.-C.C.; Visualization, C.-M.W.; Writing—original draft, C.-M.W., Y.-J.C. and S.-C.C.; Writing—review and editing, C.-M.W., Y.-J.C. and S.-C.C. All authors have read and agreed to the published version of the manuscript.

**Funding:** This research was funded by the Ministry of Science and Technology, Taiwan (https://www.most.gov.tw), grant number MOST 109-2224-E-218-001, MOST 108-2221-E-218-017-MY2, and MOST 108-2221-E-168-008-MY2 to help us finish the study successfully.

**Acknowledgments:** We hope to appreciate the financial support of the Ministry of Science and Technology, Taiwan for the projects (MOST 109-2224-E-218-001, MOST 108-2221-E-218-017-MY2, and MOST 108-2221-E-168 -008-MY2) to help us finish the study successfully. We also thank National Cheng Kung University Hospital approve this IRB project supervision (No. B-ER-105-396). Finally, we want to appreciate each member of this research team and supports from biomedical electronics center (BMEC), Southern Taiwan University of Science and Technology.

**Conflicts of Interest:** The authors declare no conflict of interest. The founding sponsors had no role in the design of the study; in the collection, analyses, or interpretation of data; in the writing of the manuscript, and in the decision to publish the results.

## Appendix A

The Morse code tables for the ACAT input mode, mouse mode, keyboard mode, and pad mode of MCT respectively can be switched from one to another according to Figure A1.

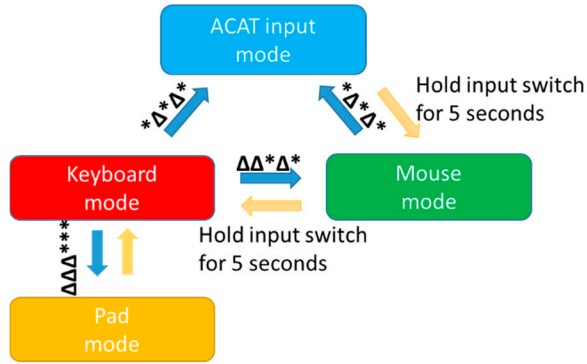

**Figure A1.** Function modes exchange of Morse code translator (MCT). (*: short tone, Δ: long tone).

When MCT works in mouse mode, the input switch must be held for 5 s continuously if the user wants to return to keyboard mode. When MCT works in ACAT input mode, the input switch must be held for 5 s continuously if the user wants to return to mouse mode. When MCT works in pad mode, the input switch must be held for 5 s continuously if the user wants to return to keyboard mode. Inversely, if MCT works in keyboard mode or mouse mode, the user can input the Morse code "*Δ*Δ*" to switch to ACAT input mode. If MCT works in keyboard mode, the user can input "ΔΔ*Δ*" to switch to mouse mode, or the user can input "ΔΔΔ***" to switch to pad mode.

**Mouse Mode Table**

| No. | Mouse Action | Code |
|-----|--------------|------|
| 1 | When the mouse is moving, stop it. | * |
| 2 | When the mouse is moving, accelerate it. | Δ |
| 3 | move right | * |
| 4 | move left | Δ |
| 5 | move up | ** |
| 6 | move down | ΔΔ |
| **7** | click left | Δ* |
| **8** | click right | *Δ |
| **9** | double click left | ΔΔ* |
| **10** | hold click left | **Δ |
| 11 | drag | ΔΔΔ |
| 12 | click middle | *** |
| 13 | move up left | **ΔΔ |
| 14 | move up right | **** |
| 15 | move down left | ΔΔΔΔ |
| 16 | move down right | ΔΔ** |
| | (*: short tone, Δ: long tone). | |

## Keyboard Mode Table

| No. | Character | Roman Pinyin | Code | No. | Character | Roman Pinyin | Code | No. | Character | Roman Pinyin | Code |
|---|---|---|---|---|---|---|---|---|---|---|---|
| 1 | A | m | *Δ | 34 | ) | | Δ**ΔΔ* | 67 | = | | *ΔΔ*Δ |
| 2 | B | r | Δ*** | 35 | 0 | an | ΔΔΔΔΔ | 68 | \<alt\> | | *Δ*ΔΔ |
| 3 | C | h | Δ*Δ* | 36 | 1 | b | *ΔΔΔΔ | 69 | \<backspace\> | | ΔΔΔΔ |
| 4 | D | k | Δ** | 37 | 2 | d | **ΔΔΔ | 70 | \<caps lock\> | | **Δ*Δ* |
| 5 | E | g | * | 38 | 3 | tone 3 | ***ΔΔ | 71 | \<ctrl\> | | Δ*Δ*Δ |
| 6 | F | q | **Δ* | 39 | 4 | tone 4 | ****Δ | 72 | \<del\> | | Δ**Δ* |
| 7 | G | sh | ΔΔ* | 40 | 5 | zh | ***** | 73 | \<end\> | | Δ*Δ** |
| 8 | H | c | **** | 41 | 6 | tone 2 | Δ**** | 74 | \<enter\> | | *Δ*Δ |
| 9 | I | o | ** | 42 | 7 | tone 0 | ΔΔ*** | 75 | \<esc\> | | **Δ** |
| 10 | J | u | *ΔΔΔ | 43 | 8 | a | ΔΔΔ** | 76 | \<home\> | | ****Δ* |
| 11 | K | e | Δ*Δ | 44 | 9 | ai | ΔΔΔΔ* | 77 | \<insert\> | | *Δ**Δ |
| 12 | L | au | *Δ** | 45 | - | er | ΔΔΔ* | 78 | \<page down\> | | ΔΔΔ*Δ* |
| 13 | M | yu | ΔΔ | 46 | ! | | *Δ**ΔΔ | 79 | \<page up\> | | ΔΔΔ**Δ |
| 14 | N | s | Δ* | 47 | # | | Δ*ΔΔΔ | 80 | \<right alt\> | | **ΔΔ*Δ |
| 15 | O | ei | ΔΔΔ | 48 | $ | | Δ***Δ* | 81 | \<right ctrl\> | | ΔΔ*Δ*Δ |
| 16 | P | en | *ΔΔ* | 49 | % | | *ΔΔ*Δ* | 82 | \<right shift\> | Chinese/English input switch for Win8, 10 | *ΔΔΔ*Δ |
| 17 | Q | p | ΔΔ*Δ | 50 | & | | Δ**ΔΔ | 83 | \<shift\> | | **Δ*Δ |
| 18 | R | j | *Δ* | 51 | * | | *Δ*** | 84 | \<space\> | | **ΔΔ |
| 19 | S | n | *** | 52 | , (comma) | eh | ΔΔ**ΔΔ | 85 | \<tab\> | | Δ*ΔΔ* |
| 20 | T | ch | Δ | 53 | . (period) | ou | *Δ*Δ*Δ | 86 | \<underline\> | | **ΔΔ* |
| 21 | U | i | **Δ | 54 | / | eng | ΔΔ**Δ | 87 | F1 | | **ΔΔΔΔ |
| 22 | V | x | ***Δ | 55 | : | | Δ*Δ*Δ* | 88 | F2 | | ***ΔΔΔ |
| 23 | W | t | *ΔΔ | 56 | ; | ang | ***Δ* | 89 | F3 | | ****ΔΔ |
| 24 | X | l | Δ**Δ | 57 | ? | | **ΔΔ** | 90 | F4 | | *****Δ |
| 25 | Y | z | Δ*ΔΔ | 58 | @ | | *ΔΔΔ* | 91 | F5 | | ****** |
| 26 | Z | f | ΔΔ** | 59 | \ | | Δ***** | 92 | F6 | | *Δ**** |
| 27 | [ | | **Δ*** | 60 | ^ | | Δ*Δ**Δ | 93 | F7 | | *ΔΔ*** |
| 28 | ] | | Δ*Δ*** | 61 | ' | | **Δ**Δ | 94 | F8 | | *ΔΔΔ** |
| 29 | { | | **ΔΔΔ* | 62 | \| | | ***Δ** | 95 | F9 | | *ΔΔΔΔ* |
| 30 | } | | Δ*ΔΔΔ* | 63 | ~ | | ΔΔ***Δ | 96 | F10 | | *ΔΔΔΔΔ |
| 31 | < | | *Δ***Δ | 64 | ' (apostrophe) | | *Δ*ΔΔ* | 97 | F11 | | Δ*ΔΔΔΔ |
| 32 | > | | ΔΔ**Δ* | 65 | " | | ΔΔ*ΔΔ | 98 | F12 | | Δ**ΔΔΔ |
| 33 | ( | | ***ΔΔ* | 66 | + | | *ΔΔ** | 99 | | | |
| | | | | | | | | | (*: short tone, Δ: long tone). | | |

## ACAT Input Mode Table

| No. | ACAT input Action | Code |
|---|---|---|
| 1 | Enter | * |
| 2 | Backspace | Δ |
| | (*: short tone, Δ: long tone). | |

**Pad Mode Table**

| No. | Pad Function | Code |
|---|---|---|
| 1 | Vol+ | * |
| 2 | Vol- | Δ |
| 3 | HOME | ** |
| 4 | Next Track | Δ* |
| 5 | Play | *Δ |
| 6 | Mute | ΔΔ |
| 7 | Stop | *** |
| 8 | Email | *Δ* |
| 9 | Search | ΔΔ* |
| 10 | Keyboard | Δ*Δ |
| 11 | Rewind | *ΔΔ |
| 12 | Browser | Δ*** |
| 13 | Fast forward | **Δ* |
| 14 | Previous Trac. | *ΔΔ* |

(*: short tone, Δ: long tone). P.S. The function of "keyboard" in pad mode table is virtual apple keyboard toggle.

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
