# Peer review of "Wireless Home Assistive System for Severely Disabled People"

_applsci, doi:10.3390/app10155226_

Round 1
Reviewer 1 Report
Authors proposed a novel assistive communication system for people with disabilities. System helps people to communicate with both other people and machine for home appliances. The system includes a Morse code translator and a PC-based augmentative and alternative communication and home automation application. System has been tested by eight severe disabled subjects.
Topic meets the journal aims and it could be useful for an always more attractive field, i.e. the domotics for people with disabilities. However, paper has to be improved, especially regarding the organization of the text.
Issues to be solved:
- Introduction: Authors should add the novelties of their system with respect the ones cited among the related works. Such information is fundamental for understanding the scientific impact of the manuscript.
- Paragraph 2.1:
- This paragraph should be reorganized since in this current version it is hard to follow. A full description of the figure 2 has to be introduced after the figure appearance and not to recall it after a lot of sentences.
- Some repetitions are present: lines 247-248 are the same of 183-184
- Line 151: what does it mean “other mechanical switches”? It is not clear if the user can add any type of mechanical switch or if there is a set of predefined switches included in the system. Similar consideration for the “and so on” referred to the analogic switches.
- Lines 165: which are the two modes? Authors reported only one repeated two times.
- The working of input accessories is often unclear maybe due to the wrong organization of this section. I suggest to firstly describe the figure 2 and then the algorithm for the Morse code translation in a sub paragraph.
- Please provide information on applied filters for the EOG signals (order, type).
- It is quite strange to have reference to the figure 7 that appears in the text after 5 pages.
- Results:
- It is not clear which was the experimental protocol that enrolled subjects have to perform. It should be described in a paragraph before the Results.
- The sentences from 328 to 340 show qualitative results not supported by any reported indices. This should be considered as a discussion, not as a result.
- Paragraph 3.1:
- I suggest moving the types of switch accessories in the paragraph 2.1
- Lines 354-370 are related to the system description and they cannot be inserted among the results. I suggest moving it into the paragraph 2.1.
- Lines 373-376 should be introduced in the previously suggested protocol section to add.
- Line 375 is a repetition of line 325.
- Paragraph 3.1.1: Lines 378-387 are related to the protocol.
- Paragraph 3.1.2: Lines 399-405 are related to the protocol.
- Paragraph 3.2: It is not clear why it is reported among the Results. It just explains some other aspects of the PC-BASED AAC.
- Paragraph 3.3: It is a repetition of line 375 and 325. Such confusion is the consequence of a lack of a protocol paragraph, in which all the details have to be declared only one time.
Author Response
To Whom This May Concern,
We much thank the reviewers for the constructive comments! Based on the review results, we were able to significantly improve the manuscript. To help the editor and reviewers go over the revision, the main responses are summarized below.
Reviewer 1:
Authors proposed a novel assistive communication system for people with disabilities. System helps people to communicate with both other people and machine for home appliances. The system includes a Morse code translator and a PC-based augmentative and alternative communication and home automation application. System has been tested by eight severe disabled subjects.
Topic meets the journal aims and it could be useful for an always more attractive field, i.e. the domotics for people with disabilities. However, paper has to be improved, especially regarding the organization of the text.
Responses:
Thank you for your valuable comments. The paper had been reorganized and the description was also improved.
Issues to be solved:
- Introduction: Authors should add the novelties of their system with respect the ones cited among the related works. Such information is fundamental for understanding the scientific impact of the manuscript.
Responses:
Thank you very much. The novelties of other researches had been added in the section “1. Introduction.” Besides, the novelties of our approach was also add in the last paragraph of section “1. Introduction.” Please refer pp. 3 and lines 105-123.
- Paragraph 2.1:
This paragraph should be reorganized since in this current version it is hard to follow. A full description of the figure 2 has to be introduced after the figure appearance and not to recall it after a lot of sentences.
Responses:
Thank you very much. In order to clearly describe the proposed approaches, the section “2. Materials and Methods” had been reorganized. Paragraph 2.1 also had been reorganized. In section 2, the Input accessories include three kinds of digital switches such as the mechanical switch, sensing switch, and bio-signal switch is introduced. Then the Morse code translator is detailed in software algorithm and hardware design. Finally, the human machine interface including PC-based AAC and home appliance control applications is mentioned. Thus, the description of figure 2 can be introduced after the figure appearance. Please refer to pp. 4 and lines 141-180.
- Some repetitions are present: lines 247-248 are the same of 183-184
Responses:
Thank you. The section “2. Materials and Methods” had been reorganized and then the repetitions had been removed.
- Line 151: what does it mean “other mechanical switches”? It is not clear if the user can add any type of mechanical switch or if there is a set of predefined switches included in the system. Similar consideration for the “and so on” referred to the analogic switches.
Responses:
Thank you for your valuable comments. The section “2. Materials and Methods” had been reorganized and then the assistive input accessories are reorganized and detailed in subsection 2.1. The input accessories can be divided into three types, consisting of mechanical, sensing, and bio-signal. All three types of input accessories mentioned above can output the similar digital outputs (different time duration of high/low-level voltages) to represent long/short tone and long/short silence of Morse codes to MCT. Any type of digital switches can be accepted as an assistive input accessory in WHAS if it can output similar digital output. Please refer to pp. 4 and lines 147-151.
- Lines 165: which are the two modes? Authors reported only one repeated two times. The working of input accessories is often unclear maybe due to the wrong organization of this section. I suggest to firstly describe the figure 2 and then the algorithm for the Morse code translation in a sub paragraph.
Responses:
Thank you very much. The section “2. Materials and Methods” had been reorganized. The two modes are not clear to represent the positive edge trigger and a negative edge trigger. Thus, the sentence had been modified as “Two types of switches were designed according to different users’ requests, one is the positive edge trigger (0→1) switch, the other is the negative edge trigger (1→0) switch.” Please refer to pp. 4 and lines 163-164.
- Please provide information on applied filters for the EOG signals (order, type).
Responses:
Thank you very much and the information of filters for the EOG signals is appended as “The 2nd order 1.6Hz~10Hz infinite impulse response filter bandpass filter is designed for noise cancellation.” Please refer to pp. 4 and line 175.
- It is quite strange to have reference to the figure 7 that appears in the text after 5 pages.
Responses:
Thanks a lot. The section “2. Materials and Methods” had been reorganized and this problem had been solved.
- Results:
It is not clear which was the experimental protocol that enrolled subjects have to perform. It should be described in a paragraph before the Results.
Responses:
Thank you very much and the enrolled subjects had been detailed in the first paragraph of Results. For each experiment, the experimental protocol was also detailed before the results. Please refer to pp. 9 and lines 326-334.
- The sentences from 328 to 340 show qualitative results not supported by any reported indices. This should be considered as a discussion, not as a result.
Responses:
Thank you very much and the qualitative results are a discussion of the proposed WHAS. Therefore, it had been moved to subsection 3.2.2. Please refer to pp. 12 and lines 447-457.
- Paragraph 3.1:
I suggest moving the types of switch accessories in the paragraph 2.1
Lines 354-370 are related to the system description and they cannot be inserted among the results. I suggest moving it into the paragraph 2.1.
Lines 373-376 should be introduced in the previously suggested protocol section to add.
Line 375 is a repetition of line 325.
Paragraph 3.1.1: Lines 378-387 are related to the protocol.
Paragraph 3.1.2: Lines 399-405 are related to the protocol.
Paragraph 3.2: It is not clear why it is reported among the Results. It just explains some other aspects of the PC-BASED AAC.
Paragraph 3.3: It is a repetition of line 375 and 325. Such confusion is the consequence of a lack of a protocol paragraph, in which all the details have to be declared only one time.
Responses:
Thank you very much. According to these comments, the manuscript had been modified as the suggests and then section 2 and 3 were reorganized
Reviewer 2 Report
With interest we were reading about the many technical devices and were impressed by the many technical interfaces. But we have the following problems:
- From the perspective of a user centered approach we miss a basic analysis of what disabled people want and consider as helpful and useful. Now the designers took the decisions about interfaces.
- The paper has no user appreciation study. Only the technical aspects are tested.
- Many similar systems have been designed and implemented. For example many wheel chair applications and voice based interfaces of home environments and device interfaces. The current interface is not benchmarked with other systems.
- The current interface is based on Morse code. It is questionable, at least not tested if Morse code is a suitable communication interface for disabled people. In many applications icons are used for communication. The researcher Stephen Hawkins also used a sech based interface because he wants to communicate on a high level. Most disabled people are happy with simple communication and then the question is if a language based interface is most suitable. At our university a wheelchair interface has been developed by a student suffering ALS disease. This interface is based on the frequency of and intensity of some basic sounds. He was able to emulate the basic functionality without copying speech commands.
- Many test results are not presented in details
- The authors present a graphic language in LabVIEW without details. The authors designed three kinds of dialogues. But these dialogues are not presented or tested
Author Response
To Whom This May Concern,
We much thank the reviewers for the constructive comments! Based on the review results, we were able to significantly improve the manuscript. To help the editor and reviewers go over the revision, the main responses are summarized below.
Reviewer 2:
With interest we were reading about the many technical devices and were impressed by the many technical interfaces. But we have the following problems:
- From the perspective of a user centered approach we miss a basic analysis of what disabled people want and consider as helpful and useful. Now the designers took the decisions about interfaces.
Responses:
Thank you very much. The analysis of disabled people was inserted in the first paragraph of section 3. In our experiments, six subjects, one subject, and one subject are SCI, CP, and ALS, respectively. Moreover, the degree of the subjects’ barriers is also detailed in Table 1. Please refer to pp. 9 and lines 326-333.
- The paper has no user appreciation study. Only the technical aspects are tested.
Responses:
Thanks very much. To evaluate the performance of proposed approaches, the subjects are asked to give the mean opinion scores (MOS) after using our proposed human machine interface in acceptance of human machine interface and degree of practicality. Please refer to pp. 12 and line 437-446.
- Many similar systems have been designed and implemented. For example many wheel chair applications and voice based interfaces of home environments and device interfaces. The current interface is not benchmarked with other systems.
Responses:
Thanks. In this study, the enrolled subjects had been analyzed and the speaking ability of them are almost faint. Besides, all of them have lied on the bed for many years. Therefore, many similar systems are not suitable for enrolled subjects. To clearly describe these conditions, the analysis of enrolled subjects is adding in the first paragraph of section 3. Please refer to pp.9 and line 326-333.
- The current interface is based on Morse code. It is questionable, at least not tested if Morse code is a suitable communication interface for disabled people. In many applications icons are used for communication. The researcher Stephen Hawkins also used a speech based interface because he wants to communicate on a high level. Most disabled people are happy with simple communication and then the question is if a language based interface is most suitable. At our university a wheelchair interface has been developed by a student suffering ALS disease. This interface is based on the frequency of and intensity of some basic sounds. He was able to emulate the basic functionality without copying speech commands.
Responses:
Thank you for your valuable comments. In this study, the speaking ability of the enrolled subjects is faint and then they cannot use speech commands to communicate with others or devices. Moreover, the subjects are finger fretting or quadriplegic, thus they are unable to control the AAC. Thus, the Morse code is one of the successful approaches to ease them in communicating with others or devices. These reasons are also added in the first paragraph of section 3. Please refer to pp. 9 and line 326-333.
- Many test results are not presented in details
Responses:
Thanks a lot. Section 3 was reorganized and then the test results are presented in detail. Please refer to pp. 12 and line 437-446.
- The authors present a graphic language in LabVIEW without details. The authors designed three kinds of dialogues. But these dialogues are not presented or tested
Responses:
Thank you very much. These three kinds of dialogues had been detailed in the last paragraph of subsection 2.4. Moreover, these human machine interfaces were examined by uses. Thus, the subjects are asked to give the mean opinion scores (MOS) to evaluate the acceptance of the human machine interface and the degree of practicality. The experimental results were added in subsection 3.2.3.
Round 2
Reviewer 1 Report
Authors improved the manuscript thanks to the suggestions of the two Reviewers. The readability of the manuscript has been improved thanks to the reorganization of some paragraphs.
Before the publication, I strongly suggest to introduce a paragraph "Experimental protocol" in which all the tasks required to the subjects enrolled in the study must be declared. Actually, the tasks are yet reported among the results and it is not common for a rigorous scientific journal.
Author Response
To Whom This May Concern,
We much thank the reviewers for the constructive comments! Based on the review results, we were able to significantly improve the manuscript. To help the editor and reviewers go over the revision, the main responses are summarized below.
Reviewer 1:
- Authors improved the manuscript thanks to the suggestions of the two Reviewers. The readability of the manuscript has been improved thanks to the reorganization of some paragraphs.
Responses:
Thank you very much.
- Before the publication, I strongly suggest to introduce a paragraph "Experimental protocol" in which all the tasks required to the subjects enrolled in the study must be declared. Actually, the tasks are yet reported among the results and it is not common for a rigorous scientific journal.
Responses:
Thank you for your valuable comments. In this study, many experiments were designed to evaluate the proposed approaches. However, the experimental protocols for Morse code translator (MCT used as mouse mode and keyboard mode) and human machine interface (PC-based AAC and home appliance control) are quite different. Therefore, it is very difficult for readers to understand the experiments when the experimental protocols are written in a paragraph. Thus, we still mentioned the experimental protocols in each subsections. Moreover, to help readers in understanding the experiments for evaluating the performance of human machine interface, the experimental protocol was added in pp. 12 and lines 439 to 441.
“In these experiments, the subjects were asked to operate the human machine interface including PC-based AAC and home appliance control by using the specific input accessories connected with MCT in two months and lasting 2 to 3 hours every day. After two months, the acceptance of user friendly and the degree of practicality are evaluated and the results are shown in Table 4.”
Reviewer 2 Report
Dear Authors,
Many thanks for the improvements and success with further research
Author Response
To Whom This May Concern,
We much thank the reviewers for the constructive comments! Based on the review results, we were able to significantly improve the manuscript. To help the editor and reviewers go over the revision, the main responses are summarized below.
Reviewer 2:
- Many thanks for the improvements and success with further research
Responses:
Thank you very much.